# Research effort devoted to regulating and supporting ecosystem services by environmental scientists and economists

Andrew N. Kadykalo[1,2,3,4]* , Lisa A. Kelly[3‡], Albana Berberi[3,4‡], Jessica L. Reid[3,4‡], C. Scott Findlay[1,2]

**1** Department of Biology, University of Ottawa, Ottawa, Ontario, Canada, **2** Institute of the Environment, University of Ottawa, Ottawa, Ontario, Canada, **3** Department of Biology, Carleton University, Ottawa, Ontario, Canada, **4** Canadian Centre for Evidence-Based Conservation and Environmental Management, Institute of Environmental Sciences, Carleton University, Ottawa, Ontario, Canada

☯ These authors contributed equally to this work.
‡ These authors also contributed equally to this work.
* andriy.kadykalo@gmail.com

## Abstract

The economic valuation of ecosystem services in part reflects the desire to use conventional economic tools (markets and economic instruments) to conserve ecosystem services. However, for regulating and supporting ecosystem services that depend on ecosystem structure and function, estimation of economic value requires estimates of the current level of underlying ecological functions first. This primary step is in principle, the job of environmental scientists, not economists. Here, we provide a coarse-level quantitative assessment of the relationship between the research effort expended by environmental scientists (on the biophysical values) and economists (on the monetary values) on 15 different regulating and supporting services in 32 ecosystem types using peer-reviewed article hits retrieved from bibliographic databases as a measure of research effort. We find a positive, moderately strong ($r = 0.69$) correlation between research efforts in the two domains, a result that, while encouraging, is likely to reflect serendipity rather than the deliberate design of integrated environmental science-economics research programs. Our results suggest that compared to environmental science research effort economic valuation is devoted to a smaller, less diverse set of ecosystem services but a broader, more diverse, set of ecosystem types. The two domains differed more with respect to the ecosystem services that are the major focus of research effort than they did with respect to the ecosystem types of principal research interest. For example, carbon sequestration, erosion regulation, and nutrient cycling receive more relative research effort in the environmental sciences; air quality regulation in economic valuations. For both domains, cultivated areas, wetlands, and urban/semi-urban ecosystem types received relatively large research effort, while arctic and mountain tundra, cave and subterranean, cryosphere, intertidal/littoral zone, and kelp forest ecosystem types received negligible research effort. We suggest ways and means by which the field of sustainability science may be improved by the design and implementation of a searchable database of environmental science and economic valuation literature as well as a global ecosystem service research network and repository that explicitly links research on the

**Data Availability Statement:** See https://doi.org/
10.5683/SP2/JP1IMV for all the extracted data
which support these results.

**Funding:** Both ANK and CSF were supported by
Smart Prosperity Institute and the Natural Sciences
and Engineering Research Council of Canada [ANK:
PGSD2-534299-2019; CSF: RGPIN170399-2001].
The funders had no role in study design, data
collection and analysis, decision to publish, or
preparation of the manuscript.

**Competing interests:** The authors have declared
that no competing interests exist.

estimation and prediction of biophysical ecosystem functions with that of the social sciences
and other knowledge systems. These suggestions would, at least in principle, facilitate a
more efficient research agenda between economists and environmental scientists and aid
management, regulatory and judicial decision-makers.

## Introduction

People and societies depend on natural or semi-natural ecosystems that provide benefits to
support human existence and wellbeing [1–3]. Recognition of this concept is arguably as old as
humanity, but has been experienced and conceptualized in multiple ways throughout human
history [4–9]. In the current western scientific discourse, this relationship between people and
nature is often viewed through the lens of "ecosystem services" [10] or "nature's contributions
to people" (NCP) [9]. Ecosystem services (also variously referred to as "ecological goods and
services", "environmental services", "nature's services", "nature's benefits to people", etc.)
characterize a broad range of benefits conferred either directly or indirectly through the bio-
physical conditions and processes of natural or semi-natural ecosystems. The Millennium Eco-
system Assessment [1] categorized ecosystem services into four broad types: provisioning,
regulating, cultural, and supporting.

'Regulating' (e.g., air quality regulation, climate regulation and carbon sequestration, flood
control) and 'supporting' (e.g., primary productivity, biogeochemistry, nutrient cycling and
provisioning of habitat) ecosystem services can be regarded as classes of ecological processes
and functions that demonstrably contribute to human welfare [10, 11]. They "fundamentally
underpin biosphere integrity, human safety, and the [delivery] of most other ecosystem ser-
vices" [12], including 'provisioning' ecosystem services (e.g., 'material' goods or products:
medicinal plants, timber, and other raw materials) and 'cultural' ecosystem services (e.g., 'non-
material' benefits: aesthetic experience, recreation and eco-tourism, sense of place). Thus, the
production of all ecosystem services depend upon the level of contributing ecosystem pro-
cesses and functions and any realistic assessment of ecosystem services is contingent on esti-
mates of the level of associated ecological functions.

However, regulating and supporting ecosystem services are undervalued and overlooked
because the benefits they provide are complex to track [12] and accounting for them may lead
to the problem of 'double counting' [13]. For example, several recent ecosystem services classi-
fication systems have moved away from the Millennium Ecosystem Assessments' 'supporting'
ecosystem services' or have re-classified these services under the most salient categories (e.g.,
'habitat services' in The Economics of Ecosystems and Biodiversity Project (TEEB) [2]; 'regula-
tion and maintenance services' in the Common International Classification of Ecosystem Ser-
vices (CICES) [14]; 'habitat creation and maintenance' in The Intergovernmental Science-
Policy Platform on Biodiversity and Ecosystem Services (IPBES) [9]).

If ecosystem services contribute to human welfare, they potentially have economic value.
While sociocultural and biophysical valuations are important, economic valuations have domi-
nated the ecosystem services assessment and decision-making landscape [15–18]. Many deci-
sions concerning sustainable ecosystem management may be influenced by the (real or
imagined) economic value attributed to associated services–see Chapter 3 of [11] for examples
of area-based planning, regulatory decision analysis, environmental damages assessment, envi-
ronmental management, conservation instruments. The possibility exists then for using stan-
dard economic tools to evaluate these services, and economic rationales to conserve or

maintain them. If, for example, the economic value of a wetland as a source of groundwater recharge, flood control, or surface water filtration is sufficiently large, then there may be a *bona fide* economic incentive in maintaining natural wetlands rather than converting them for residential or commercial use.

The level of a regulating or supporting ecosystem service provisioning depends on the level of the ecosystem functions that sustain the service: if the level of one or more underlying functions is changed, so too will be the level of service delivery and hence, the economic value of the delivered services. Thus, any reliable quantitative estimate of the economic value of a specific regulating or supporting ecosystem service depends on having reliable quantitative estimates of the level of the ecological functions that sustain it, functions that reflect the biophysical properties of the ecosystem under scrutiny. The implication is that methods for deriving quantitative estimates of the economic value of ecosystem services are of limited value in the decision-making context without an accompanying estimate of the level of the underlying ecological functions [19–23]. Reciprocally, the value of a quantitative estimate of the level of ecosystem functioning is enhanced in a decision-making context if accompanied by a robust estimate of the economic value of the sustained ecosystem services. Consequently, there is considerable value added in an integrated research agenda whereby economists focus on ecosystems and their associated services for which (a) current biophysical scientific knowledge permits some level of prediction about the level of the associated underlying ecological functions and (b) there are tools that can be deployed to generate robust quantitative estimates of the economic value of sustained ecosystem services based–in part–on the level of underlying ecological functions.

Several scientometric analyses (measuring and analysing scientific literature) have sought to explore trends and evolution of the ecosystem services concept and literature. These reviews generally found that research topics between social and natural science disciplines on ecosystem services to be relatively fragmented. Abson et al. [24] found considerable compartmentalization of ecosystem services research between 1997 and 2011: ordination and clustering of 1,388 peer-reviewed publications revealed a distinct gradient from social science research focusing on economic valuations to natural science research dealing with biodiversity, and ecosystem functions and processes. Similarly, Chaudhary et al. [25] found that despite 'ecological economics' and 'ecology/biodiversity' composing ecosystem services subject areas/themes with the highest number of articles, approximately only 1% of all 519 analyzed articles were classified as 'integrated ecology and economics'. McDonough et al. [26] found that ecosystem services articles published between 2005 and 2016 in the Scopus database were mostly characterized as environmental sciences (34%) and economics (3%), with only a mere 1% of publications characterized as "multidisciplinary". More recently, Chan and Satterfield [22] analyzed more than 1,000 articles addressing ecosystem services published between 1990 and 2017 and concluded that despite 24% of the sample of studies being coded as 'biophysical', there is a "continued pre-occupation with numerical valuation often without appropriate biophysical grounding". Namely, they found a fraction of the representative sample conducted biophysically grounded valuation (2.4 ± 0.5%), and this fraction seemed invariant across time. While Droste et al. [27] found a recent (2011–2016) shift towards integrated assessments in ecosystem services research based on a bibliometric content analysis of 14,118 peer-reviewed abstracts, to the best of our knowledge, no one has asked the question of whether economists and environmental scientists work on the same ecosystem services in the same ecosystem types.

Here we consider this question of the empirical relationship between the research effort expended by economists and environmental scientists on different ecosystem types, and different regulating and supporting ecosystem services? where research effort is quantified by the

number of peer-reviewed article publication hits (i.e., counts) in a set of bibliographic databases. This also helps to highlight deficiencies in research effort on different ecosystem types and ecosystem services of environmental science studies and economic valuations. Bibliometric methods employing publication counts have been used as indices to provide overviews of trends in research efforts across broad fields of inquiry [28–31] in both the environmental [e.g., 32–34] and social [e.g., 35, 36] sciences. Although such metrics do not capture a number of important dimensions of research effort (e.g., including the number of researchers involved, the quality and costs of research, and the like), they nonetheless provide a crude measure that can be informative.

## Materials and methods

We conducted a scientometric analysis of the scientific literature exploring the relationship in research effort between economists and environmental scientists on ecosystem services in specific ecosystem types (i.e., whether they work on the same ecosystems and/or services).

### Methodological process and approach

We assembled a database of hits retrieved from a set of electronic bibliographic databases, employing search strings corresponding to a defined set of biophysical ecosystem services and ecosystem types. The database included: (1) the target ecosystem service and associated ecosystem process/function search terms; (2) an ecosystem type designation; and (3) the number of peer-reviewed article hits (i.e., publication counts of article 'document results') associated with a specific combination of ecosystem service and ecosystem type obtained by searching a specific database. Fifteen biophysical ecosystem services (S1 Table) and 32 ecosystem classes (whether subsystems, biome/ecoregion, or anthromes) here referred to as 'ecosystem types' (S2 Table) were selected and considered in the analysis (S1 File).

### Literature searches

Searches were systematic but not comprehensive: our aim was a representative sample of the literature in both (i.e., environmental science and economic valuation) domains of inquiry. Because we were not undertaking a systematic review or meta-analysis, articles were *not* screened for eligibility with respect to a particular scientific hypothesis. Rather, search strategies were developed to reduce the probability of a study concerned primarily with the economic valuation of a service being captured in a search for articles concerned with the biophysical characterization of ecosystem services, and *vice versa*.

**Keyword sensitivity analysis.**   Before final searches were performed keywords in search strings were evaluated for their ability to accurately identify relevant articles. To do so, we generated independent samples of hits retrieved from both Web of Science (Core Collection) and Scopus using (a) environmental science search fields and strings; (b) economic valuation search fields and strings. For a given search, the abstract for each retrieved hit was read to determine if it was indeed an environmental science study (in the case of (a)) or economic valuation study (in the case of (b)). To develop the search fields and strings, we started with an initial set of keywords based on our defined set of 15 regulating and supporting ecosystem services (S1 Table) and 32 ecosystem classes (S2 Table). Because keywords specific to identifying environmental science studies are diffuse, while keywords specific to economic valuation are more concentrated and precise, our search strategies differed for each domain of inquiry. We targeted environmental sciences through an exclusion search field and keywords (Table 1) and economic valuation studies through an inclusion search field and keywords (Table 2). See https://doi.org/10.5683/SP2/JP1IMV; Kadykalo_etal_ESRE_data_2.tab for economic valuation

**Table 1. Search fields and associated search strings for retrieving environmental science research on regulating and supporting ecosystem services from Web of Science and Scopus.**

| Search Field | Description | Search String |
|---|---|---|
| FIELD 1: | Regulating or supporting ecosystem service (processes and functions) | Literature search string from S1 Table |
| AND FIELD 2: | To capture ecosystem services-specific literature | "ecosystem service*" OR "ecological service*" OR "environmental service*" OR "nature's benefit* to people" OR "nature's contribution* to people" |
| AND NOT FIELD 3: | Exclusion string to exclude articles that may tend to relate to economic valuation, policy, social sciences and humanities | "avoided cost*" OR "avoided damage*" OR "benefit* transfer" OR "carbon market*" OR "choice experiment*" OR "contingent valu*" OR "cny" OR "cultural ecosystem service valu*" OR "ESV" OR "ESVs" OR "farmer* attitude*" OR "farmer* preference*" OR "farmer* value*" OR "green accounting" OR "hedonic pric*" OR "hedonic model*" OR "importance performance analysis" OR "local attitude*" OR "local* perception*" OR "market based instrument*" OR "monetization" OR "monetary valu*" OR "nature based solution" OR "net present value" OR "option value" OR "PES" OR "payment* for ecosystem service*" OR "payment* for environmental service*" OR "payment* for ecological service*" OR "planning discourse" OR "policy implementation*" OR "pollination market*" OR "public attitude*" OR "public preference*" OR "public perception*" OR "rancher* perception*" OR "rancher* perspective*" OR "REDD" OR "replacement cost*" OR "RMB" OR "resident* attitude*" OR "resident* perception*" OR "resident* perspective*" OR "resident* value*" OR "socio cultural valu*" OR "smallholder household*" OR "stakeholder analysis" OR "stakeholder attitude*" OR "stakeholder dialogue" OR "stakeholder perception*" OR "total economic value" OR "total value of ecosystem service*" OR "travel cost*" OR "US dollar*" OR "U.S. dollar*" OR "USD" OR "VES" OR "willing to pay" OR "willingness to pay" OR "WTP" OR "yuan" |
| AND FIELD 4: | Ecosystem type | Literature search string from S2 Table |

**Table 2. Search fields and associated search stings for retrieving economic valuation research on regulating and supporting ecosystem services from Web of Science and Scopus.**

| Search Field | Description | Search String |
|---|---|---|
| FIELD 1: | Regulating or supporting ecosystem service (processes and functions) | Literature search string from S1 Table |
| AND FIELD 2: | To capture ecosystem services-specific literature | "ecosystem service*" OR "ecological service*" OR "environmental service*" OR "nature's benefit* to people" OR "nature's contribution* to people" |
| AND FIELD 3: | To capture economic valuation specific literature | "avoided cost*" OR "avoided damage*" OR "benefit* transfer" OR "cny" OR "contingent valu*" OR "ESV" OR "ESVs" OR "hedonic pric*" OR "hedonic model*" OR "net present value" OR "replacement cost*" OR "RMB" OR "travel cost*" OR "US dollar*" OR "U.S. dollar*" OR "USD" OR "willing to pay" OR "willingness to pay" OR "WTP" OR "yuan" |
| AND NOT FIELD 4: | Exclusion string to exclude payment for ecosystem services articles | "PES" OR "payment* for ecosystem service*" OR "payment* for environmental service*" OR "payment* for ecological service*" |
| AND FIELD 5: | Ecosystem type | Literature search string from S2 Table |

keywords that were screened and reasons for inclusion or exclusion into the search string. Using relevance as the objective function, we iteratively refined the set of keywords within search strings until at least 95% of retrieved articles were deemed relevant. Thus, for each independent sample of article hits retrieved from searches, we evaluated the abstract for each article hit for relevancy to generate an estimate of the proportion of retrieved article hits that were 'accurate', i.e., the proportion of article hits using an environmental science search that were in fact about environmental science. We also recorded which keyword was employed in capturing individual relevant articles. Additional keywords from the abstract, title, or database keywords that captured articles of interest were iteratively added to search strings to inform subsequent test searches. Keywords from abstract, title, or database keywords which captured irrelevant articles were iteratively added to the *AND NOT* (exclusion) search fields. As mentioned, environmental science searches added keywords to an *AND NOT* (exclusion) string to exclude articles that were not environmental science studies (i.e., those that may tend to relate to economic valuation, policy, social sciences and humanities, etc.) (Table 1). Economic valuation searches had an *AND NOT* (exclusion) search string of its own, to exclude payment for ecosystem services articles which were selected with economic valuation keywords but generally did not include economic valuations (i.e., monetary values) (Table 2). Successive relevance assessment and adaptive iteration using independent samples of article hits resulted in a final set of economic valuation and environmental science search strings that achieved 95–100% relevance in both Web of Science (Core Collection) and Scopus (January 24-April 7, 2020). See Kadykalo_etal_ESRE_data_3.tab for results of the keyword sensitivity analysis, as well as Kadykalo_etal_ESRE_data_4.tab for the final list of synonyms (ecosystem processes and functions) used as keywords in literature search strings).

**Search process.** Since no single electronic database of scientific literature indexes all peer-reviewed literature, we used two different multidisciplinary Academic Citation Indices: Thompson Reuter's Web of Science and Elsevier's Scopus. For Web of Science, we used the 'Core Collection' exclusively as preliminary searches indicated that using any other Web of Science's databases (e.g., BIOSIS, SciELO) identified many document records that were *not* captured by the systematic search strings employed (i.e., they were selected unsystematically via predictive analytics based on proximity to one or more provided keywords). Searches were limited to academic peer-reviewed articles. In both Web of Science and Scopus, we thus limited our search results to 'articles' only (i.e., books, proceedings, reviews etc. and other 'document types' were excluded). No date or language restrictions were imposed on database searches, although search strings were exclusively in English.

Searches for environmental science research included four search fields that were searched using the 'Topic' (Web of Science) or 'Article title, Abstract, Keywords' (Scopus) field codes (Table 1). For two services, 'disease regulation' and 'water purification and waste treatment' the keyword "environmental service\*" was removed from 'Field 2' as it captured internal medicine/health care sciences and solid waste management articles, respectively.

All articles retrieved using a specific search string (that is, combination of ecosystem type and ecosystem service synonyms) were considered a hit. As there are 32 ecosystem types and 15 ecosystem services, there are $N = 480$ combinations in total, with a given database yielding a count (number of article hits) for each combination. Eliminating the fourth search field provided an index of the total absolute research effort allocated by environmental sciences to a particular regulating or supporting ecosystem service, pooled over the full set of *potential* ecosystem types. Eliminating the first search field provides an index of the total absolute research effort allocated by environmental sciences to a particular ecosystem type, pooled over the full set of *potential* ecosystem services (which may include provisioning and cultural ecosystem services). The same article could be retrieved for multiple ecosystem service/type combinations

if, for example, the level of the service was estimated for two or more ecosystem types, or multiple services were estimated for the same ecosystem type. The procedure for generating estimates of economic valuation research effort was the same, with the addition of an *AND* search field to capture economic valuation articles and a modified *AND NOT* (exclusion) search field (Table 2).

As with environmental science research, articles retrieved using all five fields resulted in a $N = 480$ ecosystem type × ecosystem service combinations, yielding a count (number of article hits in economic valuation research effort) for each combination. Elimination of the first search field provides an index of the total absolute research effort allocated to the economic valuation of a particular ecosystem type, pooled over the full set of *potential* ecosystem services (which may include provisioning and cultural ecosystem services). Elimination of the fifth search field provides an index of total absolute research effort allocated to the economic valuation of a particular regulating or supporting ecosystem service, pooled over the full set of *potential* ecosystem types. Because the search strategy for environmental sciences excluded the economic valuation search string (Field 3), the count totals for the two disciplines reflect a completely distinct set of retrieved articles for each combination of ecosystem service and ecosystem type, thereby insuring complete independence of the two samples.

## Data extraction

Final literature searches described above were conducted by 4 researchers (authors AK, LK, AB, JR) from April 16–28, 2020. Inter-rater reliability of the search results was estimated for a sample ecosystem service (carbon sequestration) for both electronic databases and research domains (number of raters = 4, number of subjects (carbon sequestration/ecosystem type combinations) = 256). For each subject, the hits retrieved by the 4 researchers were compared. Percentage agreement (for the number of retrieved article hits) among researchers (99.8%) and inter-rater reliability (Fleiss' Kappa = 0.989) were calculated using the R package 'irr' [37]. The two genuine disagreements in the number of retrieved article hits among raters were a result of recording the wrong document type (a review instead of article in Web of Science), and an error in search syntax (missing brackets) in Scopus. Differences between researchers were discussed and resolved to inform subsequent searches and data extraction.

## Data analysis

**Variation among ecosystem types and ecosystem services.**   In order to explore the variation among both ecosystem types and ecosystems services between environmental science and economic valuation literature we performed a two-factor ANOVA without replication. Eta-squared ($\eta2$), a measure of effect size, was calculated to estimate the proportion of variance associated with ecosystem type or ecosystem services effects among article hits. Eta-squared ($\eta2$) is the amount of variation explained in the outcome variable (Y) explained by the predictor variable (X), calculated as $\eta2 = \sigma2$ effect/ $\sigma2$ total, where $\sigma2$ effect is the sum of squares (SS) of the predictor and the $\sigma2$ total is the SS Total. In subsequent analyses, raw article hits were $\log_{10}+1$ transformed to reduce the large differences in average hits between the environmental sciences and economic valuation and to accommodate ecosystem type × ecosystem service combinations that had zero hits.

**Research effort differential.**   To explore the fine scale correlation between environmental science and economic valuation on individual combinations of the selected ecosystem services and types and to identify outlier combinations we calculated a research effort differential ($\log_{10}+1$ number of hits in environmental sciences–$\log_{10}+1$ number of hits in economic valuation) for each of the $N = 480$ ecosystem type × ecosystem service combinations. We then

calculated the average differential ($\bar{x} = 0.49$) and associated standard deviation (SD = ± 0.48) for each of the 480 combinations, as well as the associated z-transformed standardized differential D* = (differential for combination i—average differential)/SD), yielding 480 z-scores. These scores allow us to directly compare the two samples by eliminating the effects of scale.

## Results

See https://doi.org/10.5683/SP2/JP1IMV for the extracted data which support these results including the raw extracted data (i.e., article hits) (Kadykalo_etal_ESRE_data_1.tab).

We retrieved 6,629 and 6,682 articles on the 15 selected regulating and supporting ecosystem services from Web of Science (WoS) and Scopus respectively, most of which were obtained using the environmental science search strategy (WoS: 5,871, Scopus: 5,611) (Kadykalo_etal_ESRE_data_5.tab). Retrieved literature spanned 27 years from 1993 to 2020. As results were qualitatively similar for the two databases, here we present results for WoS; SCOPUS results are given in S2 File.

Averaged over the $N = 480$ ecosystem type × ecosystem service combinations, the number of article hits was far greater in environmental sciences (15.6 ± 40.9 (1 SD) versus 1.2 ± 2.8 (1 SD) for economic valuation). Research effort in both domains varied substantially among both ecosystem types and ecosystem services, with more ecosystem type than ecosystem service variation in the economic evaluation literature, and the converse in environmental science (Fig 1). Thus, economic valuation research effort is devoted to a smaller, less diverse set of ecosystem services (presumably those for which might have markets or economic valuation is possible) but a broader, more diverse, set of ecosystem types than is environmental science research effort.

Overall, there was a moderately strong ($r = 0.69$, Fig 2A) correlation between research effort in the two domains based on the number of article hits for each of the ecosystem type × ecosystem service combinations. Focusing solely on ecosystem services (i.e., correlation based on N = 15 ecosystem services) or ecosystem types (i.e., correlation based on N = 32 ecosystem types) improved the correlation ($r = 0.80$ (Fig 2B) and 0.88 (Fig 2C), respectively) between research efforts in the two domains. For individual ecosystem services, research effort correlations were uniformly high (range 0.72–0.92; S3 File); by contrast, ecosystem types correlations showed considerably more variation (range 0.32–0.93; S3 File).

At a gross scale, patterns of research effort in the two domains were similar. For both domains, cultivated areas, rivers, wetlands, and urban/semi-urban ecosystem types received relatively large research effort, while aquaculture, arctic tundra, mountain tundra, cave and subterranean, cryosphere, ephemeral wetland, montane grasslands and shrubland, intertidal/ littoral zone, and kelp forest ecosystem types received very little research effort (Fig 3). Although carbon sequestration, erosion regulation, water purification and waste treatment, and water regulation received considerable research effort in both domains, there were nonetheless significant differences between the two domains. Biological control, air quality regulation, and coastal and storm protection are services of intense interest (in terms of research effort) to economists in a wide range of ecosystems; whereas for environmental scientists, the principal focus is cultivated (biological control), urban/semi-urban areas (air quality regulation), or ocean ecosystems (coastal and storm protection) (Fig 3). By contrast, disease regulation and drought mitigation receive comparatively little effort by economists but are widely studied by environmental scientists.

At a finer scale, cultivated areas, streams and creeks, surface open ocean & deep sea, tropical/subtropical forests/woodlands, and urban/semi-urban ecosystem types receive relatively more research effort in the environmental sciences as compared to economic valuations (Fig

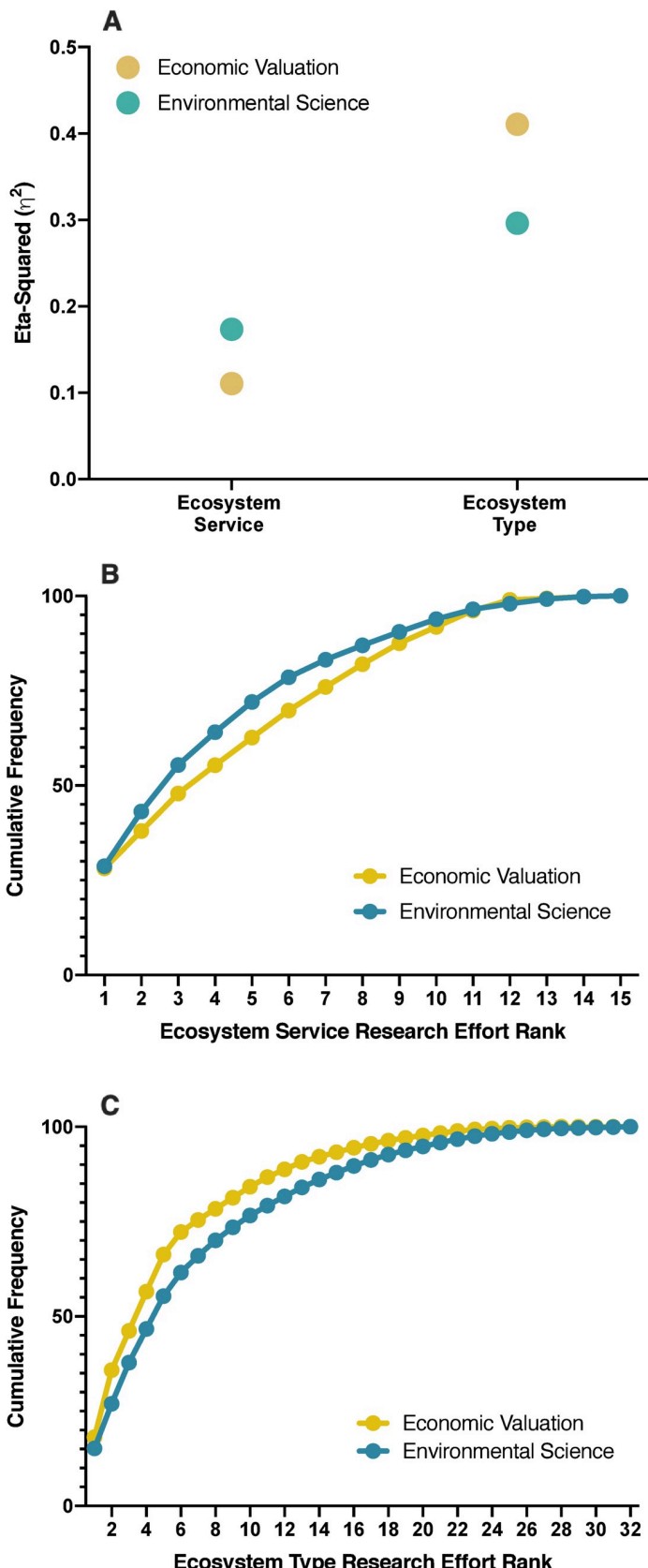

**Fig 1. Variation among ecosystem types and ecosystem services.** (A) Eta-squared ($\eta^2$) for a two-factor (ecosystem service, ecosystem type) ANOVA of the number of article hits in the two research domains. Also shown are cumulative frequency plots for $N$ = 15 ecosystem services (B) and $N$ = 32 ecosystem types (C). See Kadykalo_etal_ESRE_data_6.csv for cumulative frequency data.

4). In terms of ecosystem services, carbon sequestration, erosion regulation, and nutrient cycling receive more relative research effort in the environmental sciences; air quality regulation in economic valuations (Fig 4). Outliers (combinations of ecosystem types and ecosystem services with especially large research effort differentials) in which there is greater economic valuation research effort include air quality regulation in lakes and mangrove forests; water regulation in mangrove forests; and flood regulation in tropical/subtropical grasslands (Fig 4). Outliers in which there is greater environmental sciences research effort include biological control in cultivated areas; carbon sequestration in seagrasses and temperate/boreal forests/woodlands; pollination in cultivated areas and tropical/subtropical forests/woodlands; seed dispersal in tropical/subtropical forests/woodlands; and soil formation in tropical/subtropical forests/woodlands.

## Discussion

Our analysis of the environmental science literature on regulating and supporting ecosystem services, and the economics literature on the valuation of associated ecosystem services, shows a moderately strong overall correlation between research efforts (as estimated by number of article hits in literature searches) in the domains of inquiry for ecosystem services (pooled over ecosystem types) and ecosystem types (pooled over services). The implication is that at a gross scale, the services and ecosystems of most or least interest to economists are also, in general, the most or least interest to environmental scientists.

The demand for and access to ecosystem services (see Chan and Satterfield [22]) underly the observed positive correlations, which would be welcome from the decision-maker's point of view. For example, doing field research in many of the under-studied ecosystems (e.g., caves, open ocean, and the arctic) is expensive and technically challenging. As a consequence, comparatively few environmental scientists have the resources and technical infrastructure to conduct research in these ecosystems compared to, for example, temperate forests. Economists would also be expected to be less interested in such ecosystems simply because these are systems in which few people live: as such, the number of people who derive benefits from ecosystem services is small, and the potential for value conflicts in managing ecosystems (e.g., development vs. preservation) reduced.

Another possibility is that when deciding upon which ecosystem services to (e)valuate and in which ecosystems to do so, economists tend to focus on those ecosystems and services for which there is a reasonable corpus of environmental science knowledge, that is, a biophysical evidence base from which to work. It seems evident that estimating the economic value of an arbitrary unit of ecosystem service is of limited value unless one also has some idea of the number of units delivered under alternative ecosystem states or management regimes, which in turn requires some understanding of the level of the biophysical functions underlying the service(s) of interest. This would serve to heighten the interest of economists in ecosystems and ecosystem services for which there already exists a substantive biophysical science literature. Consistent with this explanation, for our sample of ecosystems and services, the associated environmental science literature (average publication year 2008 ± 7.36) was generally older than the corresponding economic valuation literature (2010 ± 6.21), likely influencing and setting trends in economic valuation research or other domains of inquiry. See also Braat and de

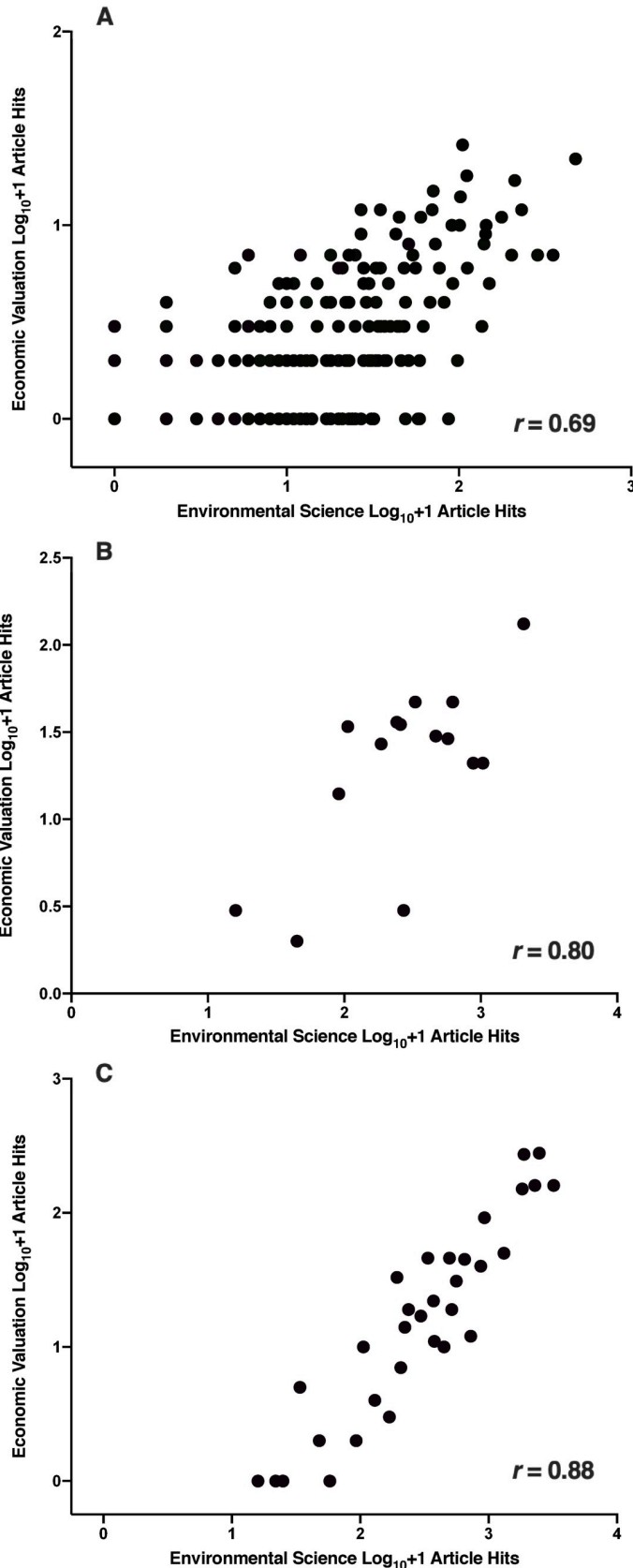

**Fig 2. Correlation between environmental science and economic valuation article hits.** Scatterplot between environmental science and economic valuation research effort as estimated by the $\log_{10}+1$ number retrieved article hits for $N = 15$ regulating and supporting ecosystem services × $N = 32$ ecosystem types (A), $N = 15$ individual regulating and supporting ecosystem services (B), and $N = 32$ individual ecosystem types (C).

Groot [38], which summarize the history of the ecosystem services concept and its ecological roots; and Droste et al. [27], which found that five out of the nine ecosystem services topics between 1990 and 2000 ('early research') to deal mainly with ecology and land use. These explanations would induce the comparatively strong correlations. Much like Chan and Satterfield [22] and McDonough et al. [26], the highest proportion of studies in our sample were in the environmental sciences; although as the former study authors contend, this proportion may still to be too rare given that biophysical change is fundamental to ecosystem services research.

These general patterns notwithstanding, there are substantial differences at the level of (ecosystem type × ecosystem service) combinations, with economic valuations tending to focus on a narrow range of ecosystem services across a broader range of ecosystems. Conversely, environmental scientists tend to focus on a broader range of ecosystem services in a comparatively

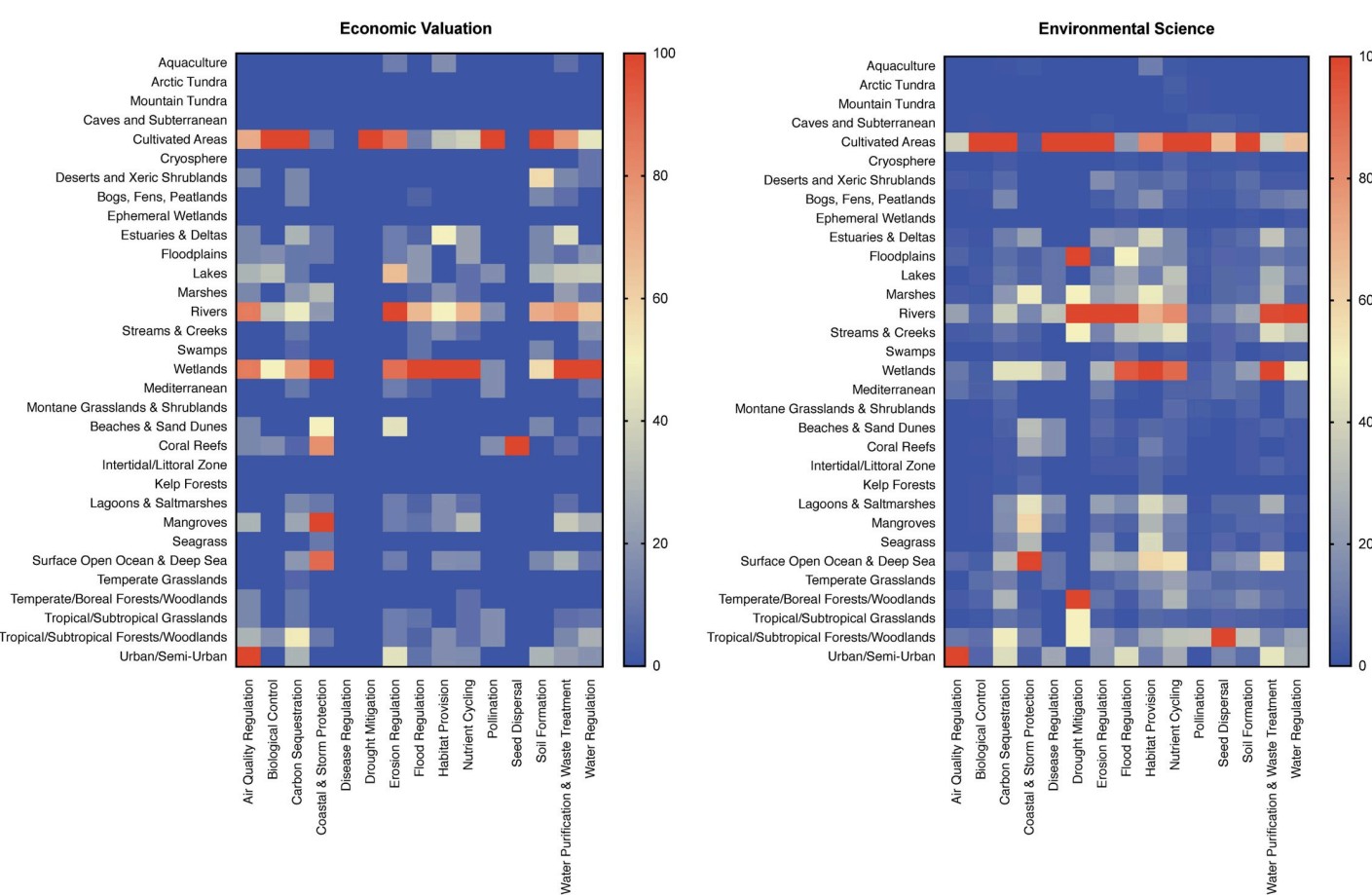

**Fig 3. Research effort in environmental science and economic valuation.** Heat map of economic valuation and environmental science research effort on each of $N = 15$ regulating and supporting ecosystem service and $N = 32$ ecosystem type combinations as estimated by retrieved article hits. Raw article hits were normalized based on the smallest (0%) and largest (100%) values in each data set to allow for direct and relative comparison between research domains with respect to relative article hits. Red cells indicate higher, blue cells indicate lower research effort for that combination of ecosystem service and ecosystem type relative to the average effort.

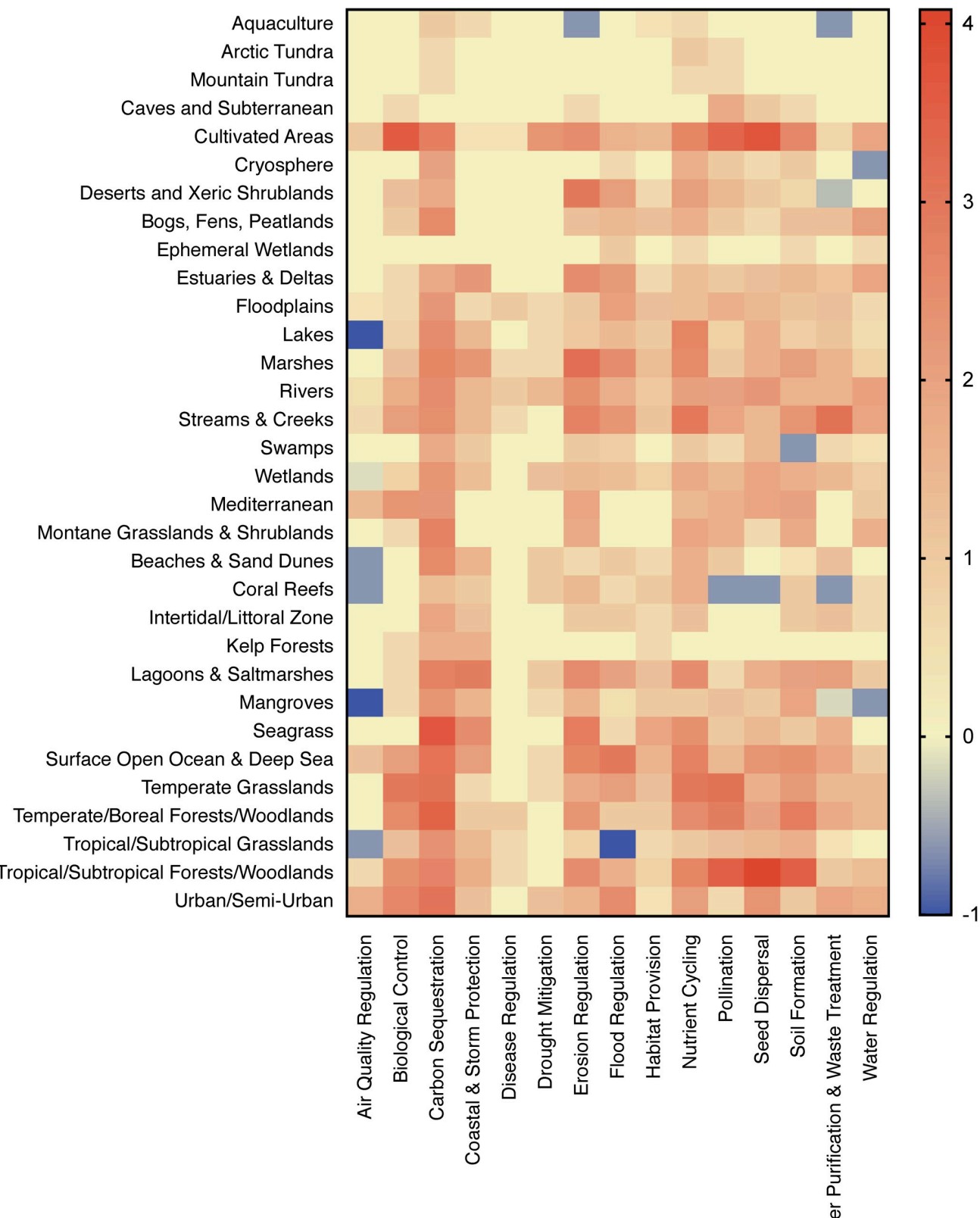

**Fig 4. Research effort differential between environmental science and economic valuation.** Heat map of the z-transformed $Log_{10}+1$ research effort differential between environmental science and economic valuation research effort on each of $N = 15$ regulating and supporting ecosystem service and $N = 32$ ecosystem type combinations as estimated by retrieved article hits. Z-transformed differentials were normalized using the z-transformed value of the raw differential of zero (-1.03) to allow for direct comparison between research domains with respect to relative article hits. Red cells indicate higher environmental sciences, blue cells indicate higher economic valuation research effort for that combination of ecosystem service relative to the average effort. See Kadykalo_etal_ESRE_data_7.tab for research effort differential data.

narrow range of ecosystems. This presents issues if, as Abson et al. [24] and Chan and Satter-field [22] have found, interdisciplinary research has relatively little focus on ecological process or functions and specifically, if economic valuations are very rarely biophysically grounded.

Our results suggest economists tend to focus on ecosystem services that have a direct link to human welfare and security in ecosystems where population density is comparatively large (and where clear marketed commodities exist). Conversely, environmental scientists tend to place a stronger focus on services with direct links to ecological functions and processes (e.g., carbon sequestration, nutrient cycling, seed dispersal, biological control and pollination). Like Droste et al. [27], we found these services are analyzed in several ecosystems of key focus (e.g., cultivated areas, freshwater, oceanic, forests). Because economists value services (e.g., biological control air quality regulation, and storm protection) in a wide range of ecosystems this indicates that economists may frequently assess ecosystem services in bundles (i.e., across multiple services and/or multiple ecosystem types, through perhaps benefits transfer or contingent valuation), whereas environmental scientists more frequently assess specific services in specific ecosystems. Similarly, Chan and Satterfield [22] found a large share of studies taking a 'total' valuation approach despite repeated calls for 'marginal' valuation based on realistic biophysical changes [e.g., 19, 39], which is more relevant for on-the-ground decision-making.

Overall, there was an overwhelming focus on human-altered/dominated ecosystem types (e.g., cultivated and urban/semi-urban areas) which is unsurprising given the anthropocentric focus of ecosystem services. This supports results by Droste et al. [27] and Chan and Satterfield [22], which found that agriculture areas (where services like pest control and pollination are quantified in both domains) are where social and natural science converge most. By contrast, several natural ecosystems with relatively high cover of the earth's surface received little research effort regardless of domain, especially those ecosystems in high elevations, polar and subterranean regions.

## Study limitations and sources of error

The estimated correlation of research effort in the two disciplines for a given ecosystem service is surprisingly strong, given the potential sources of error. One obvious source of error is the difficulty in completely and comprehensively characterizing a specific ecosystem service with respect to a defined set of search phrases and strings. If the probability of retrieving a relevant article, given a particular search phrase, differs between the two disciplines, this will, in effect, increase measurement error, thereby reducing the observed correlation. Additionally, the potential overlap between several ecosystem services (i.e., lack of fine distinctions in cases between water purification & waste treatment and nutrient cycling; flood control and water regulation) is also likely to have the same effect. However, we tested our ability to detect studies that perform biophysical or economic valuation by screening a sample of the collected articles to verify relevance and adherence to our search strings. We note another limitation, that studies that estimate ecosystem services across large areas are unlikely to contain all the regulating and supporting ecosystem services or ecosystem types in the abstract, title, and keywords–a limitation applying to other reviews of abstracts [e.g., 22, 26, 27]. Other limitations also include a bias towards English language articles, which is a similar limitation of others [e.g., 24–27].

Research effort on lakes, oceans, and rivers may also be inflated due to these terms being commonly used to describe 'places' rather than references to research within true ecosystem types. Notwithstanding these study limitations and sources of error, the detected correlations between economist and environmental scientist research efforts are comparatively and surprising strong, suggesting that indeed, the actual correlations are even stronger.

Future investigation on the research effort devoted to ecosystem services (or other topics) could increase the sophistication and detail of this analysis by including other dimensions of research effort. The number of researchers, the rigour of methods applied, the transparency with which the methods are reported, the limitations imposed in the research, the quantity and quality of the primary data, and the costs and funding of scientific research enterprises may be important indicators of research effort. Such study limitations could be addressed in future research through the use of interviews or surveys of researcher or funders; datamining of funding awards by granting agencies (e.g., National Science Foundation, Canada's Tri-Agency Financial Administration, dimensions.ai); and of course, full-length article assessment using an assessment tool (e.g., The Collaboration for Environmental Evidence Synthesis Appraisal Tool, see Konno et al. [40]).

## Implications to ecosystem management and decision-making

Our results indicate there are many ecosystems for which there is both a corpus of environmental science and economic valuation work. This suggests that in the context of environmental decision-making, a reasonable presumption is that there exists a corpus of relevant economic valuation work that can—and indeed should—be drawn upon. For example, under Canada's *Impact Assessment Act* (IAA), where contribution to sustainability is an explicit determinative public interest factor (see IAA S.C. 2019, s.63), regulators should adopt the presumption that there may well exist a relevant literature on the economic value of ecosystem functions that might be affected by the proposed project. Project proponents should therefore be expected to demonstrate that they have made efforts to track down and employ relevant findings in assessing the economic impacts of the project arising from effects on ecosystem services. In the legal context, this knowledge could also be used to provide valuable information about the associated economic costs of environmental damage resulting from regulatory offences, with economic cost estimates in particular helping courts to determine appropriate remedies, sanctions or punishments.

## Searchable database of environmental science and economic valuation literature

In both instances cited above, a determination of (a) the relevant economic valuation literature and (b) the extent to which it should inform (management, regulatory or judicial) decisions would be expedited by a tool (searchable database) which provides a searchable and current repository of environmental science and economic valuation literature, referenced by ecosystem service and ecosystem type. Such a tool would either complement or build upon the TEEB Ecosystem Valuation Database [16, 41–44] and the Environmental Valuation Refence Inventory (EVRI) [45]. Both databases are currently, however, exclusively concerned with the economic valuation literature with no explicit links to the corresponding literature in the environmental sciences.

## Global ecosystem service research network and repository

Although our analysis suggests strong correlations in research effort, these correlations exist at scales that are, in general, incommensurate with the scale of ecosystem management and

decision-making. For decision-makers, the issue is: what are the consequences of decision A versus B to delivery of ecosystem service X, and the associated economic implications, in this particular geographical region over some defined timescale? Answering this question to the required level of spatiotemporal resolution requires a coordinated/integrated assessment of (a) the predicted effects of a candidate decision on a defined set of ecosystem functions (and associated services) and (b) an economic valuation of these services at the spatiotemporal scales defined by the decision context. While our results indicate that, at least at a coarse level, environmental scientists and economists are pulling in the same direction, our results provide little insight into the extent to which environmental scientists and economists are engaged in joint research enterprises at the spatiotemporal scales relevant to decision-making.

Though integrated research enterprises involving both social and environmental scientists are desirable [22, 46–51], our findings of substantial differences at the level of (ecosystem type × ecosystem service) combinations confirms observations of other authors, which suggest such enterprises still appear to be lacking [24, 25, 52, 53]. Yet it is precisely this enterprise that has, at present, the best chance of developing and deploying a set of robust tools for assigning (biophysical, economic, and even socio-cultural) value to ecosystem services—that, we believe, is required to achieve the full promise of the ecosystem service or NCP approach.

In our view, this ambitious agenda would be facilitated by the establishment of a global ecosystem service research network and repository that would complement the recently launched EcoService Models Library (https://esml.epa.gov/), aimed at linking ecological functions and processes to ecosystem services via ecological production functions [54, 55]. Further, a transparent and searchable global repository would explicitly link environmental science researchers whose expertise includes the estimation and prediction of ecosystem functions with researchers from the social sciences and other, i.e., non-western knowledge systems and sources (i.e., Indigenous and local knowledge) on the valuation of ecosystem services via three separate major link attributes: (a) geospatial and temporal location; (b) ecosystem type(s) and; (c) ecosystem services and associated ecosystem functions. Such a repository would also permit researchers in the biophysical sciences to link directly with researchers interested in the development and application of methods to estimate the economic value of such services. Finally, such a network would, at least in principle, facilitate a more efficient research agenda whereby (a) economists focus on ecosystems and their associated services for which the current scientific knowledge permits some level of prediction about, minimally, current service levels; and (b) environmental scientists focus on ecosystems and their associated services for which validated economic valuation tools exist and can be deployed. With the integration of these concepts, we are cautiously hopeful that the 'Ecosystem Services Partnership' (ESP) (https://www.es-partnership.org/) or 'Biodiversity and Ecosystem Services Network' (Bes-Net) (https://www.besnet.world/), which already have the necessary network foundations (e.g., members and working groups), can facilitate the development of such a repository and network.

## Conclusion

Overall, we found a positive, moderately strong correlation between research efforts on ecosystem services and ecosystems in the environmental science and economic valuation domains, a result that, while encouraging, is likely to reflect serendipity rather than the deliberate construction of integrated environmental science-economics research programs. Indeed, at finer scales of analysis, some combinations of ecosystem types and ecosystem services demonstrated large research effort differentials and some ecosystem services and ecosystems received relatively no research regardless of domain. Future research in both domains should attempt to

increase effort in those ecosystem services and ecosystems for which there are clearly literature gaps as identified in our analysis. Finally, although there is evidence that integrated assessments on ecosystem services and ecosystems are becoming more common, the design and implementation of a searchable database of environmental science and economic valuation literature as well as a global ecosystem service research network and repository would, at least in principle, facilitate a more efficient research agenda between economists and environmental scientists and aid management, regulatory and judicial decision-makers.

## Supporting information

**S1 Table. List of 15 regulating and supporting ecosystem services.** Biophysical ecosystem processes and functions described as regulating and supporting ecosystem services [a](Value of Nature to Canadians Study Taskforce 2017), regulating NCP categories [b](Díaz et al. 2018), and their corresponding synonyms (processes and functions) used as keywords in a literature search strings in bibliographic searches.
(PDF)

**S2 Table. List of 32 ecosystem classes (whether subsystems, biome/ecoregion, or anthromes) here referred to as 'ecosystem types'.** Ecosystems as broad 'biomes' or 'ecoregions' and 'anthromes' classes, which were further refined into finer-scale 'subsystems' and the corresponding keywords used in literature search strings in bibliographic searches for ecosystem type. Due to the diversity of inland wetland types, we employed both a general wetland search terms as well as terms based on higher resolution wetland classes.
(PDF)

**S1 File. Selection of biophysical ecosystem services and ecosystem types.** Description of how regulating and supporting ecosystem services as well as ecosystem types were selected.
(PDF)

**S2 File. Scopus-specific results.** The Number of Article Hits–Scopus; S2.1 Fig in S2 File. Variation among ecosystem types and ecosystem services (Scopus); Correlational Analysis–Scopus (S2.2 Fig in S2 File. Correlation between environmental science and economic valuation article hits); S2.3 Fig in S2 File. Research effort in environmental science and economic valuation (Scopus); S2.4 Fig in S2 File. Research effort differential between environmental science and economic valuation (Scopus).
(PDF)

**S3 File. Additional results.** Correlational Analysis–Web of Science vs. Scopus; Fig 3.1 in S3 File. Correlation between environmental science and economic valuation article hits for each of the 15 biophysical ecosystem services; S3.1 Table in S3 File. Pearson correlation coefficients between economic valuation and environmental science research effort on 15 selected biophysical ecosystem services; S3.2 Table in S3 File. Pearson correlation coefficients between economic valuation and environmental science research effort on 32 selected ecosystem types; S3.3 Table in S3 File. The total and mean number of retrieved article hits in Web of Science and Scopus for environmental science and economic valuation research effort on 15 biophysical ecosystem services; S3.2 Fig in S3 File. Bar graph of economic valuation and environmental science research effort on $N = 15$ selected biophysical ecosystem services, pooling over all potential ecosystem types; S3.4 Table in S3 File. The total and mean number of retrieved article hits in Web of Science and Scopus for environmental science and economic valuation research effort on 32 ecosystem types; S3.3 Fig in S3 File. Bar graph of economic valuation and environmental science research effort on each of $N = 32$ ecosystem types, pooling over all potential

ecosystem services).
(PDF)

## Acknowledgments

We thank Angela Zheng, Paulina Pisarek, Kayla Grey, and Jake Hendrick for preliminary data collection. We thank Heather MacDonald for advice around systematic database searching and Tammy Newcomer-Johnson for a discussion on US EPA's EcoService Models Library. We thank four anonymous referees for commenting thoughtfully on our manuscript.

## Author Contributions

**Conceptualization:** Andrew N. Kadykalo, C. Scott Findlay.

**Data curation:** Andrew N. Kadykalo, Lisa A. Kelly, Albana Berberi, Jessica L. Reid.

**Formal analysis:** Andrew N. Kadykalo, Lisa A. Kelly, Albana Berberi, Jessica L. Reid.

**Funding acquisition:** Andrew N. Kadykalo, C. Scott Findlay.

**Investigation:** Andrew N. Kadykalo, Lisa A. Kelly, Albana Berberi, Jessica L. Reid, C. Scott Findlay.

**Methodology:** Andrew N. Kadykalo, C. Scott Findlay.

**Project administration:** Andrew N. Kadykalo, C. Scott Findlay.

**Resources:** Andrew N. Kadykalo, C. Scott Findlay.

**Software:** Andrew N. Kadykalo.

**Supervision:** Andrew N. Kadykalo, C. Scott Findlay.

**Validation:** Andrew N. Kadykalo.

**Visualization:** Andrew N. Kadykalo.

**Writing – original draft:** Andrew N. Kadykalo.

**Writing – review & editing:** Andrew N. Kadykalo, Lisa A. Kelly, Albana Berberi, Jessica L. Reid, C. Scott Findlay.

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
