## [Decision Letter · Decision Letter 0]

26 Mar 2021

PONE-D-21-00613

Research effort devoted to regulating and supporting ecosystem services by environmental scientists and economists

PLOS ONE

Dear Dr. Kadykalo,

Thank you for submitting your manuscript to PLOS ONE. After careful consideration, we feel that it has merit but does not fully meet PLOS ONE’s publication criteria as it currently stands. Therefore, we invite you to submit a revised version of the manuscript that addresses the points raised during the review process.

In your revision please address the comments/suggestions made by both reviewers and in particular clarify the comments made by reviewer #2 regarding adding some clarification on the literature search performed.

We look forward to receiving your revised manuscript.

Kind regards,

Andrea Belgrano, Ph.D.

Academic Editor

PLOS ONE

Journal Requirements:

Reviewers' comments:

Reviewer's Responses to Questions

**Comments to the Author**

1. Is the manuscript technically sound, and do the data support the conclusions?

Reviewer #1: No

Reviewer #2: Yes

2. Has the statistical analysis been performed appropriately and rigorously? 

Reviewer #1: Yes

Reviewer #2: Yes

3. Have the authors made all data underlying the findings in their manuscript fully available?

Reviewer #1: Yes

Reviewer #2: Yes

4. Is the manuscript presented in an intelligible fashion and written in standard English?

Reviewer #1: Yes

Reviewer #2: Yes

5. Review Comments to the Author

Reviewer #1: The comments and detailed review report are included in the attached document. Unconveniently, the system seems to require me to include them here again.

Review report on manuscript PONE D-21-00613

“Research effort devoted to regulating and supporting ecosystem services by environmental scientists and economists”

The paper presents a bibliographical analysis of research done in the domain of ecosystem services, focusing on environmental science aspects on the one hand, and on economic valuation, on the other. For the main part, the paper contrasts the frequencies

at which different types of ecosystems and different types of ecosystem services have been studied in one or the other field. As the literature is expanding, an attempt to quantitatively analyze and structure this literature is in principle welcome. Yet, the analysis in the present paper remains largely descriptive, and provides little analytical insight.

In the following I list my specific questions, comments, and concerns.

1. I do not agree that valuation always requires to quantify ecosystem services in physical units. For the so-called ‘cultural ecosystem services’ this is quite obvious. For example, the value of recreational benefit can be measured in in monetary units, whereas I have no idea how environmental scientists could physically quantify recreational services without somehow resorting to values. As another example, tn the list of NCPs there is the ‘maintenance of options’. This is also one example where valuation is possible, but physical quantification hard to imagine.

Against this background, a solid argument is needed why physical quantification is a prerequisite for valuation when it comes to regulating and supporting ecosystem services. As it stands, this is merely an unsupported claim that is repeatedly made

in the paper.

2. That said, it is required to know how ecosystems are functioning if the aim is to make a valuation study useful for decision-making. Otherwise it would not be possible to assess how management changes would change the value of ecosystem services. So my previous remark does not invalidate the claim that an integrated research agenda is desirable.

3. The measure of research effort devoted to certain types of ecosystem services or types of ecosystems should somehow be normalized. One of the reasons why cultivated areas are more researched than kelp forest ecosystems probably is the much larger area covered by the former.

4. For this particular paper it might be useful to mention the disciplinary background of the authors: How many of them are economists, how many are environmental scientists?

5. I do not generally object the search strings applied. However, more needs to be said on the method how they were generated. As such, the choice of the search string merely seems to appeal to plausibility.

6. Figure 2. Why is the data not presented on a linear scale?

7. Figures 3/4. I do not easily get the analytical message that these heat maps are supposed to convey. The seem to be purely descriptive to me.

8. Is there any evidence that supports the hypotheses why economists and environmental scientists choose their respective focus?

9. The implications for ecosystem management and decision-making seem not supported by the analysis. In particular the fact that “there exists a corpus of relevant economic valuation work” was known before.

10. I agree that the “results provide little insight into the extent to which environmental scientists and economists are engaged in joint research enterprises”. Thus, also no conclusions should be drawn in this regard.

11. The correlation of research in environmental science and valuation may be driven by societal demand, which should be welcome from the decision-maker’s point of view.

Reviewer #2: An important issue in developing policy for global environmental stewardship is having reliable economic valuations of ecosystem of ecosystem services, globally. The degree to which this is possible rests on having reliable environmental scientific insights about levels of ecosystem functions and services across the wide range of ecosystems on Earth. This MS reports on an analysis aimed at discerning the breadth of coverage of scientific insights about ecosystem services and whether or not that breadth of insight aligns with the breadth of coverage of economic valuations. This is an interesting analysis that helps to highlight deficiencies in the scope of coverage of both environmental science insights and environmental economic valuations. The MS is clearly written.

The study essential draws associations between the number of studies conducting research on 15 different services x 32 ecosystems and the number of studies conducting economic analyses of each 15 x 32 combination of function and ecosystem type. The data are gathered by conducting a search of both the environmental scientific and environmental economic literature using electronic databases. Care is taken to ensure that the articles selected from the environmental science literature is independent of the literature collection from the environmental economic valuation literature. The search procedure is for the most well explained and hence repeatable. The data are appropriately analyzed.

I only have a few questions to help with clarification.

In the description of the literature search, it was unclear how an article was deemed acceptable. Was it based on the its title? Or was there a further reading of the Abstract or article text to gain insights about what the study did? The reason I ask is that often article titles can misrepresent article contents, so that, for example, what may seem to be an environmental scientific assessment ends up being merely a call for environmental science assessments or a discussion of what should be done to assess the level of function or service. The same could be said for economic valuations. Please explain how the search process verified that scientific measurements or economic valuations were actually done in a given study.

The search was conducted by four individuals. The methods describe a test for agreement among the four about whether an article should be included or not. But how did the search procedure avoid duplication of article selection among the four searchers, which could otherwise inflate the samples size for each service x ecosystem type category? Pleas eexplain.

Line 230: in the estimation of Eta-squared, it is not clear what sigma-effect and sigma-total are and how they are calculated. Please explain more.

Line 241-246: what is the rationale for doing a log transformation?

6. PLOS authors have the option to publish the peer review history of their article (what does this mean?). If published, this will include your full peer review and any attached files.

Reviewer #1: No

Reviewer #2: No

---

## [Author Response · Author response to Decision Letter 0]

30 Apr 2021

Response to Reviewers

Reviewer #1: The comments and detailed review report are included in the attached document.

Unconveniently, the system seems to require me to include them here again. 

“Research effort devoted to regulating and supporting ecosystem services by environmental

scientists and economists”

The paper presents a bibliographical analysis of research done in the domain of ecosystem

services, focusing on environmental science aspects on the one hand, and on economic

valuation, on the other. For the main part, the paper contrasts the frequencies

at which different types of ecosystems and different types of ecosystem services have been

studied in one or the other field. As the literature is expanding, an attempt to quantitatively analyze and structure this literature is in principle welcome. Yet, the analysis in the present paper remains largely descriptive, and provides little analytical insight.

Response: Thank you for your support in our approach and objective. Moreover, thank you for the care in providing specific comments in helping us improve the paper. We argue that this paper does provide analytical insight by a) measuring the variation in research effort in ecosystem services and types, b) estimating the correlational relationship between domains of inquiry, and c) measuring the research effort differential to analyze for which ecosystem services and ecosystem type combinations, which domain has more or less relative research. We elaborate on this in responses to specific comments below.

In the following I list my specific questions, comments, and concerns.

1. I do not agree that valuation always requires to quantify ecosystem services in physical units.

For the so-called ‘cultural ecosystem servicesʼ this is quite obvious. For example, the value of

recreational benefit can be measured in in monetary units, whereas I have no idea how

environmental scientists could physically quantify recreational services without somehow

resorting to values. As another example, tn the list of NCPs there is the ‘maintenance of optionsʼ.

This is also one example where valuation is possible, but physical quantification hard to imagine.

Against this background, a solid argument is needed why physical quantification is a

prerequisite for valuation when it comes to regulating and supporting ecosystem services. As it

stands, this is merely an unsupported claim that is repeatedly made in the paper.

2. That said, it is required to know how ecosystems are functioning if the aim is to make a

valuation study useful for decision-making. Otherwise it would not be possible to assess how

management changes would change the value of ecosystem services. So my previous remark

does not invalidate the claim that an integrated research agenda is desirable.

Response to 1 and 2: Thank you your comment which allows us to clarify. We agree that biophysical quantification is not a prerequisite for the valuation of all ecosystem services. We have attempted to clarify that we are speaking specifically to regulating and supporting ecosystem services that depend on ecosystem structure and function. We have made the following changes, in the abstract: “However, for regulating and supporting ecosystem services that depend on ecosystem structure and function, estimation of economic value requires estimates of the current level of underlying ecological functions first” and in the introduction: “The level of a regulating or supporting ecosystem service provisioning depends on the level of the ecosystem functions that sustain the service: if the level of one or more underlying functions is changed, so too will be the level of service delivery and hence, the economic value of the delivered services. Thus, any reliable quantitative estimate of the economic value of a specific regulating or supporting ecosystem service depends on having reliable quantitative estimates of the level of the ecological functions that sustain it, functions that reflect the biophysical properties of the ecosystem under scrutiny. The implication is that methods for deriving quantitative estimates of the economic value of ecosystem services are of limited value in the decision-making context without an accompanying estimate of the level of the underlying ecological functions [19-23]. Reciprocally, the value of a quantitative estimate of the level of ecosystem functioning is enhanced in a decision-making context if accompanied by a robust estimate of the economic value of the sustained ecosystem services. Consequently, there is considerable value added in an integrated research agenda whereby economists focus on ecosystems and their associated services for which (a) current biophysical scientific knowledge permits some level of prediction about the level of the associated underlying ecological functions and (b) there are tools that can be deployed to generate robust quantitative estimates of the economic value of sustained ecosystem services based – in part – on the level of underlying ecological functions.”

Comment: 3. The measure of research effort devoted to certain types of ecosystem services or types of ecosystems should somehow be normalized. One of the reasons why cultivated areas are more researched than kelp forest ecosystems probably is the much larger area covered by the former.

Response: Yes, this is the reason for the log transformation. Apologies that we did not provide a rational previously. Due to the large difference in article hits between domains of inquiry (see results; average article hits were dramatically greater in the environmental sciences), raw article hits were log10+1 transformed to reduce this skew, improving linearity between our dependent and independent variables. Because many searches resulted in 0 article hits and a log of 0 is undefined, we took the log10 of (the article hits +1).

New text reads, “In subsequent analyses, raw article hits were log10+1 transformed to reduce the large differences in average hits between the environmental sciences and economic valuation and to accommodate ecosystem type × ecosystem service combinations that had zero hits.”

z-transformation allows us to directly compare the two samples by eliminating the effects of scale. 

New text reads, “To explore the fine scale correlation between environmental science and economic valuation on individual combinations of the selected ecosystem services and types and to identify outlier combinations we calculated a research effort differential (log10+1 number of hits in environmental sciences – log10+1 number of hits in economic valuation) for each of the N = 480 ecosystem type × ecosystem service combinations. We then calculated the average differential (x̄ = 0.49) and associated standard deviation (SD = ± 0.48) for each of the 480 combinations, as well as the associated z-transformed standardized differential D* = (differential for combination i - average differential)/SD), yielding 480 z-scores. These scores allow us to directly compare the two samples by eliminating the effects of scale.”

Comment: 4. For this particular paper it might be useful to mention the disciplinary background of the authors: How many of them are economists, how many are environmental scientists?

Response: Most of us are interdisciplinary scientists who use both quantitative and qualitative methodologies to integrate social, economic, and ecological data and thus feel our disciplinary backgrounds have little influence on biasing one discipline over the other or bearing on how the paper should be interpreted by the reader.

Comment: 5. I do not generally object the search strings applied. However, more needs to be said on the method how they were generated. As such, the choice of the search string merely seems to appeal to plausibility.

Response: Apologies for the confusion and thanks for commenting that this is unclear. We have moved the section of ‘Keyword sensitivity analysis’ before the ‘search process’ section as this is the correct order in which they proceeded, and it may help clarify some of the concerns. We also made some substantial edits in the attempt to improve clarity. We also included links to the data in which keywords were selected. Keywords were analyzed for their ability to detect relevant articles. Keywords were iteratively added or excluded based on whether they detected an environmental science (or economic valuation) or not. After keywords in search strings achieved “95% relevance,” that is the proportion of articles that were an environmental science study, these keywords were ‘approved’ for final searches in which no screening was performed – that is, only article hits were recorded. We hope this is clearer now.

The new section now reads, 

“Keyword sensitivity analysis

Before final searches were performed keywords in search strings were evaluated for their ability to accurately identify relevant articles. To do so, we generated independent samples of hits retrieved from both Web of Science (Core Collection) and Scopus using (a) environmental science search fields and strings; (b) economic valuation search fields and strings. For a given search, the abstract for each retrieved hit was read to determine if it was indeed an environmental science study (in the case of (a)) or economic valuation study (in the case of (b)). To develop the search fields and strings, we started with an initial set of keywords based on our defined set of 15 regulating and supporting ecosystem services (S1 Table) and 32 ecosystem classes (S2 Table). Because keywords specific to identifying environmental science studies are diffuse, while keywords specific to economic valuation are more concentrated and precise, our search strategies differed for each domain of inquiry. We targeted environmental sciences through an exclusion search field and keywords (Table 1) and economic valuation studies through an inclusion search field and keywords (Table 2). See https://doi.org/10.5683/SP2/JP1IMV; Kadykalo_etal_ESRE_data_2.tab for economic valuation keywords that were screened and reasons for inclusion or exclusion into the search string. Using relevance as the objective function, we iteratively refined the set of keywords within search strings until at least 95% of retrieved articles were deemed relevant. Thus, for each independent sample of article hits retrieved from searches, we evaluated the abstract for each article hit for relevancy to generate an estimate of the proportion of retrieved article hits that were ‘accurate’, i.e., the proportion of article hits using an environmental science search that were in fact about environmental science. We also recorded which keyword was employed in capturing individual relevant articles. Additional keywords from the abstract, title, or database keywords that captured articles of interest were iteratively added to search strings to inform subsequent test searches. Keywords from abstract, title, or database keywords which captured irrelevant articles were iteratively added to the AND NOT (exclusion) search fields. As mentioned, environmental science searches added keywords to an AND NOT (exclusion) string to exclude articles that were not environmental science studies (i.e., those that may tend to relate to economic valuation, policy, social sciences and humanities, etc.) (Table 1). Economic valuation searches had an AND NOT (exclusion) search string of its own, to exclude payment for ecosystem services articles which were selected with economic valuation keywords but generally did not include economic valuations (i.e., monetary values) (Table 2). Successive relevance assessment and adaptive iteration using independent samples of article hits resulted in a final set of economic valuation and environmental science search strings that achieved 95-100% relevance in both Web of Science (Core Collection) and Scopus (January 24-April 7, 2020). See Kadykalo_etal_ESRE_data_3.tab for results of the keyword sensitivity analysis, as well as Kadykalo_etal_ESRE_data_4.tab for the final list of synonyms (ecosystem processes and functions) used as keywords in literature search strings).”

Comment: 6. Figure 2. Why is the data not presented on a linear scale?

Response: As above, apologies that we did not provide a rational previously. Due to the large difference in article hits between domains of inquiry (see results; average article hits were dramatically greater in the environmental sciences), raw article hits were log10+1 transformed to reduce this skew, improving linearity between our dependent and independent variables. Because many searches resulted in 0 article hits and a log of 0 is undefined, we took the log10 of (the article hits +1).

New text reads, “In subsequent analyses, raw article hits were log10+1 transformed to reduce the large differences in average hits between the environmental sciences and economic valuation and to accommodate ecosystem type × ecosystem service combinations that had zero hits.”

Comment: 7. Figures 3/4. I do not easily get the analytical message that these heat maps are supposed to convey. The seem to be purely descriptive to me.

Response: Thanks for commenting that the analytical message is not clear. We have revised the captions to indicate that they provide analysis that permits direct relative research effort between domains of inquiry. We have also provided details on colours to allow for clearer interpretation. 

Captions now read,

“Fig 3. Research effort in environmental science and economic valuation. Heat map of economic valuation and environmental science research effort on each of N = 15 regulating and supporting ecosystem service and N = 32 ecosystem type combinations as estimated by retrieved article hits. Raw article hits were normalized based on the smallest (0%) and largest (100%) values in each data set to allow for direct and relative comparison between research domains with respect to relative article hits. Red cells indicate higher, blue cells indicate lower research effort for that combination of ecosystem service and ecosystem type relative to the average effort.

Fig 4. Research effort differential between environmental science and economic valuation. Heat map of the z-transformed Log10+1 research effort differential between environmental science and economic valuation research effort on each of N = 15 regulating and supporting ecosystem service and N = 32 ecosystem type combinations as estimated by retrieved article hits. Z-transformed differentials were normalized using the z-transformed value of the raw differential of zero (-1.03) to allow for direct comparison between research domains with respect to relative article hits. Red cells indicate higher environmental sciences, blue cells indicate higher economic valuation research effort for that combination of ecosystem service relative to the average effort. See Kadykalo_etal_ESRE_data_7.tab for research effort differential data.”

Comment: 8. Is there any evidence that supports the hypotheses why economists and environmental scientists choose their respective focus?

Response: We ran a literature search but could find no such evidence. We believe this part of the novelty of the contribution, mapping which ecosystem services and types are the focus of economists and environmental scientists.

Comment: 9. The implications for ecosystem management and decision-making seem not supported by the analysis. In particular the fact that “there exists a corpus of relevant economic valuation work” was known before.

Response: We slightly disagree. The degree to which the average decision-maker making ecosystem management decisions considers economic valuation is likely quite low. We believe that that perhaps in the environmental economics community that “there exists a corpus of relevant economic valuation work” was known before, but not outside of it. We have revised the first sentence of our implications section to read, “Our results indicate there are many ecosystems for which there is both a corpus of environmental science and economic valuation work”.

Comment: 10. I agree that the “results provide little insight into the extent to which environmental scientists and economists are engaged in joint research enterprises”. Thus, also no conclusions should be drawn in this regard.

Response: Thank you, we agree. We have deleted the following sentence: “However, the promise and progress of ‘sustainability science’ [47-48] would seem to depend upon an integrated research enterprise involving both social and environmental scientists.”

Comment: 11. The correlation of research in environmental science and valuation may be driven by societal demand, which should be welcome from the decision-makerʼs point of view. 

Response: We agree. That is why we already had a topic sentence on this, but we added the second half of your comment: “The demand for and access to ecosystem services (see Chan and Satterfield [22]) underly the observed positive correlations, which would be welcome from the decision-maker’s point of view.”

Reviewer #2: An important issue in developing policy for global environmental stewardship is

having reliable economic valuations of ecosystem of ecosystem services, globally. The degree

to which this is possible rests on having reliable environmental scientific insights about levels of

ecosystem functions and services across the wide range of ecosystems on Earth. This MS

reports on an analysis aimed at discerning the breadth of coverage of scientific insights about

ecosystem services and whether or not that breadth of insight aligns with the breadth of

coverage of economic valuations. This is an interesting analysis that helps to highlight

deficiencies in the scope of coverage of both environmental science insights and environmental

economic valuations. The MS is clearly written.

The study essential draws associations between the number of studies conducting research on

15 different services x 32 ecosystems and the number of studies conducting economic analyses

of each 15 x 32 combination of function and ecosystem type. The data are gathered by

conducting a search of both the environmental scientific and environmental economic literature

using electronic databases. Care is taken to ensure that the articles selected from the

environmental science literature is independent of the literature collection from the

environmental economic valuation literature. The search procedure is for the most well

explained and hence repeatable. The data are appropriately analyzed.

Response: Thank you for your strong support of this work and care in providing helpful edits.

I only have a few questions to help with clarification.

Comment: In the description of the literature search, it was unclear how an article was deemed acceptable. Was it based on the its title? Or was there a further reading of the Abstract or article text to gain insights about what the study did? The reason I ask is that often article titles can misrepresent article contents, so that, for example, what may seem to be an environmental scientific assessment ends up being merely a call for environmental science assessments or a discussion of what should be done to assess the level of function or service. The same could be said for economic valuations. Please explain how the search process verified that scientific

measurements or economic valuations were actually done in a given study.

Response: Apologies for the confusion and thanks for commenting that this is unclear. We have moved the section of ‘Keyword sensitivity analysis’ before the ‘search process’ section as this is the correct order in which they proceeded, and it may help clarify some of the concerns. We also made some substantial edits in the attempt to improve clarity. Keywords were analyzed for their ability to detect relevant articles. Articles were then screened based on the abstract. Keywords were iteratively added or excluded based on whether they detected an environmental science (or economic valuation) or not. After keywords in search strings achieved “95% relevance,” that is the proportion of articles that were an environmental science study, these keywords were ‘approved’ for final searches in which no screening was performed – that is, only article hits were recorded. We hope this is clearer now.

The new section now reads, 

“Keyword sensitivity analysis

Before final searches were performed keywords in search strings were evaluated for their ability to accurately identify relevant articles. To do so, we generated independent samples of hits retrieved from both Web of Science (Core Collection) and Scopus using (a) environmental science search fields and strings; (b) economic valuation search fields and strings. For a given search, the abstract for each retrieved hit was read to determine if it was indeed an environmental science study (in the case of (a)) or economic valuation study (in the case of (b)). To develop the search fields and strings, we started with an initial set of keywords based on our defined set of 15 regulating and supporting ecosystem services (S1 Table) and 32 ecosystem classes (S2 Table). Because keywords specific to identifying environmental science studies are diffuse, while keywords specific to economic valuation are more concentrated and precise, our search strategies differed for each domain of inquiry. We targeted environmental sciences through an exclusion search field and keywords (Table 1) and economic valuation studies through an inclusion search field and keywords (Table 2). See https://doi.org/10.5683/SP2/JP1IMV; Kadykalo_etal_ESRE_data_2.tab for economic valuation keywords that were screened and reasons for inclusion or exclusion into the search string. Using relevance as the objective function, we iteratively refined the set of keywords within search strings until at least 95% of retrieved articles were deemed relevant. Thus, for each independent sample of article hits retrieved from searches, we evaluated the abstract for each article hit for relevancy to generate an estimate of the proportion of retrieved article hits that were ‘accurate’, i.e., the proportion of article hits using an environmental science search that were in fact about environmental science. We also recorded which keyword was employed in capturing individual relevant articles. Additional keywords from the abstract, title, or database keywords that captured articles of interest were iteratively added to search strings to inform subsequent test searches. Keywords from abstract, title, or database keywords which captured irrelevant articles were iteratively added to the AND NOT (exclusion) search fields. As mentioned, environmental science searches added keywords to an AND NOT (exclusion) string to exclude articles that were not environmental science studies (i.e., those that may tend to relate to economic valuation, policy, social sciences and humanities, etc.) (Table 1). Economic valuation searches had an AND NOT (exclusion) search string of its own, to exclude payment for ecosystem services articles which were selected with economic valuation keywords but generally did not include economic valuations (i.e., monetary values) (Table 2). Successive relevance assessment and adaptive iteration using independent samples of article hits resulted in a final set of economic valuation and environmental science search strings that achieved 95-100% relevance in both Web of Science (Core Collection) and Scopus (January 24-April 7, 2020). See Kadykalo_etal_ESRE_data_3.tab for results of the keyword sensitivity analysis, as well as Kadykalo_etal_ESRE_data_4.tab for the final list of synonyms (ecosystem processes and functions) used as keywords in literature search strings).”

Comment: The search was conducted by four individuals. The methods describe a test for agreement among the four about whether an article should be included or not. But how did the search procedure avoid duplication of article selection among the four searchers, which could

otherwise inflate the samples size for each service x ecosystem type category? Pleas eexplain. 

Response: We hope the above comment, that we have re-ordered the methods (‘keyword sensitivity analysis’ where articles were screened, followed by the ‘search process’ where they were not) provides better clarification. The test of agreement was about the number of retrieved article hits, not whether an article should be included or not. We have made some edits in attempt to make this clearer.

It now reads,

“For each subject, the hits retrieved by the 4 researchers were compared. Percentage agreement (for the number of retrieved article hits) among researchers (99.8%) and inter-rater reliability (Fleiss’ Kappa = 0.989) were calculated using the R package ‘irr’ [38]. The two genuine disagreements in the number of retrieved article hits among raters were a result of recording the wrong document type (a review instead of article in Web of Science), and an error in search syntax (missing brackets) in Scopus. Differences between researchers were discussed and resolved to inform subsequent searches and data extraction.”

Comment: Line 230: in the estimation of Eta-squared, it is not clear what sigma-effect and sigma-total are and how they are calculated. Please explain more.

Response: We have provided an additional sentence to explain how eta-squared is calculated and what it measures, “Eta-squared (η2) is the amount of variation explained in the outcome variable (Y) explained by the predictor variable (X), calculated as η2 = σ2 effect/ σ2 total, where σ2 effect is the sum of squares (SS) of the predictor and the σ2 total is the SS Total.”

Comment: Line 241-246: what is the rationale for doing a log transformation?

Response: Thanks for noticing that we did not provide a rationale for the log transformation. Due to the large difference in article hits between domains of inquiry (see results; average article hits were dramatically greater in the environmental sciences), raw article hits were log10+1 transformed to reduce this skew, improving linearity between our dependent and independent variables. Because many searches resulted in 0 article hits and a log of 0 is undefined, we took the log10 of (the article hits +1).

New text reads, “In subsequent analyses, raw article hits were log10+1 transformed to reduce the large differences in average hits between the environmental sciences and economic valuation and to accommodate ecosystem type × ecosystem service combinations that had zero hits.”

---

## [Editor Report · Decision Letter 1]

17 May 2021

Research effort devoted to regulating and supporting ecosystem services by environmental scientists and economists

PONE-D-21-00613R1

Dear Dr. Kadykalo,

We’re pleased to inform you that your manuscript has been judged scientifically suitable for publication and will be formally accepted for publication once it meets all outstanding technical requirements.

Kind regards,

Andrea Belgrano, Ph.D.

Academic Editor

PLOS ONE

---

## [Editor Report · Acceptance letter]

19 May 2021

PONE-D-21-00613R1 

Research effort devoted to regulating and supporting ecosystem services by environmental scientists and economists 

Dear Dr. Kadykalo:

I'm pleased to inform you that your manuscript has been deemed suitable for publication in PLOS ONE. Congratulations! Your manuscript is now with our production department. 

Kind regards, 

on behalf of

Dr. Andrea Belgrano 

Academic Editor

PLOS ONE